# Dysbiotic Gut Bacteria in Obesity: An Overview of the Metabolic Mechanisms and Therapeutic Perspectives of Next-Generation Probiotics

**DOI:** 10.3390/microorganisms10020452

**Published:** 2022-02-16

**Authors:** Jonathan Breton, Marie Galmiche, Pierre Déchelotte

**Affiliations:** 1Inserm UMR 1073, Nutrition, Inflammation and Microbiota-Gut-Brain Axis (ADEN) Laboratory, 76183 Rouen, France; jonathan.breton1@univ-rouen.fr (J.B.); pierre.dechelotte@chu-rouen.fr (P.D.); 2Institute for Research and Innovation in Biomedicine (IRIB), University of Rouen Normandy, 76000 Rouen, France; 3Department of Nutrition, CHU Rouen, U 1073, Normandie University, UNIROUEN, 76000 Rouen, France

**Keywords:** gut microbiota, obesity, next-generation probiotics, metabolic syndrome, intestinal dysbiosis

## Abstract

Obesity, a worldwide health concern with a constantly rising prevalence, is a multifactorial chronic disease associated with a wide range of physiological disruptions, including energy imbalance, central appetite and food reward dysregulation, and hormonal alterations and gut dysbiosis. The gut microbiome is a well-recognized factor in the pathophysiology of obesity, and its influence on host physiology has been extensively investigated over the last decade. This review highlights the mechanisms by which gut dysbiosis can contribute to the pathophysiology of obesity. In particular, we discuss gut microbiota’s contribution to host energy homeostatic changes, low-grade inflammation, and regulation of fat deposition and bile acid metabolism via bacterial metabolites, such as short-chain fatty acids, and bacterial components, such as lipopolysaccharides, among others. Finally, therapeutic strategies based on next-generation probiotics aiming to re-shape the intestinal microbiota and reverse metabolic alterations associated with obesity are described.

## 1. Introduction

Obesity is a worldwide health concern with a constantly rising prevalence [1]. Indeed, according to the WHO, 39% of the world’s adult population were overweight in 2016 and 13% were obese. The prevalence obesity likewise rose from 7% in 1980 to 12.5% in 2015, representing an almost 80% increase [2]. Obesity is associated with increased cardio-metabolic risk, including type 2 diabetes (T2D) and low-grade inflammation of inflated fat stores. Obesity is characterized by an energy imbalance resulting from several factors, including central appetite and food reward signaling dysregulation, and multiple biological, histological, immunological, and metabolic changes in adipose, hepatic, muscular, brain, and intestinal tissues [3]. While early reports revealed the presence of gut dysbiosis in patients with obesity, efforts over the last decade have concentrated on elucidating bacterial signaling pathways that may influence host physiology and induce or maintain obesity. Indeed, gut bacteria are implicated in the regulation of a wide range of physiological and pathophysiological processes, such as adiposity, homeostasis, inflammation, and insulin resistance, via the production of various microbial metabolites and components, peptides, and proteins. Thus, a shift from the healthy symbiosis between the microbiota and the host to persistent dysbiosis may contribute to excess fat deposition and resultant complications.

Although much evidence highlights alterations in the composition of the gut microbiota in obese patients, the functional significance of the dysbiotic intestinal bacteria needs to be summarized. Therefore, this narrative review, based on scientific articles from electronic databases (Pubmed, Google Scholar), focuses on the main mechanisms associated with the gut microbiota’s contributions to the pathophysiology of obesity and the treatment possibilities associated with these different mechanisms. Specifically, we discuss the contribution of gut microbiota to the host energy homeostasis, low-grade inflammation, and regulation of fat deposition and bile acid metabolism via the bacterial production of bacterial metabolites, such as short-chain fatty acids (SCFAs), and bacterial components, such as lipopolysaccharides (LPS), for instance. We then described the new therapeutic next-generation probiotic strategies currently being developed to re-shape the intestinal microbiota and reverse metabolic alterations promoted by obesity.

## 2. Metabolic Mechanism Linking Gut Microbiota and Obesity

### 2.1. Obesity-Associated Dysbiosis

The gut microbiome is composed of trillions of microorganisms (including fungi, yeast bacteriophages, all outcompeted by bacteria) interacting with the host physiology. Advanced technologies, called omics (i.e., metagenomics, meta-transcriptomics, meta-proteomics, and metabolomics) are now available to decipher in depth this interaction by analyzing the bacterial genes/products extracted from fecal samples or intestinal biopsies. Technical progress in the study of the gut microbiome allowed us to identify higher bacterial diversity in healthy individuals [4].

Alteration of the gut microbiota composition might play a strong role in obesity pathophysiology. Ley and colleagues identified via 16S rRNA gene sequencing a lower abundance of Bacteroidetes phylum and a significant increase of Firmicutes levels in a leptin-deficient (ob/ob) obese mouse model [5]. Few months later, Turnbaugh, from the same team, confirmed the increased Firmicutes vs. Bacteroidetes ratio of cecal bacterial DNA of this obese murine model as compared to lean healthy mice via the shotgun metagenomics sequencing technique. Ob/ob mice also exhibited higher levels of Archaea within the cecal microbial community as compared to control mice [6]. These modifications of the bacterial abundance have led to deeper gut microbial investigations in other obesity models and in humans. Therefore, other obesity-related studies have shown an association with an increased abundance of specific bacteria, such as Halomonas or Sphingomonas, and a reduced abundance of Bifidobacteria [7].

Although the composition of the gut microbiota is relatively diverse in healthy individuals, those exhibiting high adiposity, insulin resistance, and dyslipidemia (which characterize obese patients) are associated with a low bacterial gene count [4], meaning a relatively poor gut microbiota. A reduced proportion of Bacteroidetes and higher levels of Firmicutes have also been observed in obese patients (see Figure 1) [5,8,9].

For instance, it has been shown that individuals with a Firmicutes/Bacteroidetes ratio of ≥1 were 23% more likely to be overweight than those with a Firmicutes/Bacteroidetes ratio of <1 [10]. However, this has not been specifically observed in other studies [11,12]. It is also worth noting that thinking in terms of bacterial phyla alone (i.e., the Firmicutes/Bacteroidetes ratio) is largely imprecise. Indeed, Firmicutes, including Clostridium, Lactobacillus, or Ruminococcus, are increased in obesity, whereas *Faecalibacterium prausnitzii* (one of the most abundant bacteria belonging to the Firmicutes phylum in the healthy human intestine) is decreased [13].

Even in specific bacterial genera, such as Lactobacillus, discrepancies are observed. For instance, Million and colleagues observed decreased levels of *Lactobacillus paracasei* (*L. paracasei*) and *Akkermansia muciniphila* (*A. muciniphila*) and increased levels of *Lactobacillus reuteri* (*L. reuteri*) and *Lactobacillus gasseri* (*L. gasseri*) in the stool of obese as compared to lean subjects [14]. Besides, Lactobacillus and Clostridium species have been implicated differentially in relation to insulin resistance in women [15]. Clostridium has been negatively associated with fasting glucose and glycated hemoglobin (HbA1c) levels, whereas the Lactobacillus abundance displayed positive correlations with these parameters [15]. Altogether, these results suggest specific involvement of the gut bacteria in the onset or perpetuation of obesity.

Despite increasing research efforts, no specific bacterial signature has been identified in obesity. Countries, food habits, physical activity, and the techniques used to investigate the gut microbial composition may reflect the heterogeneity of the findings observed in the studied populations.

Of note, many studies have described rapid modifications of the gut microbial composition following a change in diet [16]. Moreover, the practice of moderate physical activity could also induce beneficial modifications in the gut microbiota, contributing to better mental health [17,18]. The gut microbiota also varies with age, with decreased diversity, which may result in bias in clinical studies [11,19]. The use of antibiotics during childhood may also represent a significant risk factor for the development of obesity (HR 1.26) since antibiotic treatment has been associated with weight gain in the first 2 years of life [20,21]. Moreover, a high-fat diet and dysbiosis observed during obesity contribute to an impairment in mucus production and an enrichment in barrier-disrupting species and a decrease in the expression of the cystic fibrosis transmembrane receptor (Ctfr) gene in mouse ileal enterocytes, causing a reduction in the mucus density, leading to increased intestinal permeability [22]. Therefore, a wide range of factors can modulate the composition of the intestinal microbiota and consequently lead to significant analysis biases. In addition, the techniques used for studying the microbiota (qPCR, 16s rRNA sequencing, shotgun metagenomics) have greatly evolved and may constitute extrinsic factors that contribute to the heterogeneity of the results observed [23]. To limit bias as much as possible, it appears essential to precisely clarify the phenotype of obese patients in clinical studies (context, lifestyle, anthropometry, medication, and comorbidities).

Nevertheless, although perfectible, these studies highlight the role of intestinal bacteria in the pathogenesis of obesity. Mechanisms by which gut microbiota may contribute to the pathophysiology of obesity are discussed below.

### 2.2. Production of Short-Chain Fatty Acids (SCFAs)

In the human gastrointestinal tract, the highest density of bacteria is located in the colon. The main sources of carbon and energy are represented by carbohydrates and proteins from food components that are undigested in the upper part of the gut [24]. The bioconversion of these different substrates by the gut microbiota involves the existence of a range of metabolic bacterial activities leading to an energy supply and the production of metabolites. The latter, in direct contact with the host cells, can influence physiological processes locally in the gut and also at the systemic level after the absorption of bacterial metabolites and their distribution to other organs. Thus, they may have an important role in the host metabolic phenotype and could contribute to the risk factors of diverse pathologies, such as obesity.

Carbohydrates that reach the colon are mainly represented by polysaccharides from cereals, fruits, and vegetables, and amount 40 to 60 g per day [25]. Gut bacteria are responsible for breaking down non-digestible polysaccharides, which is also called fiber. Microbes, likely anaerobes in the gastrointestinal tract, are capable of fermentation, thanks to a set of hydrolytic enzymes, which end carbohydrate conversion by producing metabolites, such as SCFAs [26]. The fibrolytic bacterial community includes species belonging to the Bacteroides, Roseburia, Ruminococcus, Bifidobacterium, Lactobacillus, and Eubacterium genera [27]. Most colonic bacterial species are characterized by fermentative activity and consequently by the production of SCFAs, such as acetate (for instance, produced by Bifidobacteria [27]), butyrate (produced by some species of Firmicutes, i.e., *Faecalibacterium prausnitzii*, *Roseburia* spp., *Eubacterium rectale* and *hallii* [28,29,30,31]), and propionate (for instance, produced Bacteroides, Prevotella, and Veillonella [32,33]). Thus, the type of SCFA depends on the composition of the microbial communities [34]. As recently reviewed, obese gut microbiota are known to have greater quantities of catabolic genes, suggesting a higher ability to extract energy from the diet (and thus to produce more SCFAs) than non-obese subjects [35,36]. Experimental studies in both obese human and animal subjects suggested that an increased production of SCFAs could provide additional calories to the host and thus, could lead to weight gain [37].

### 2.3. The Role of SCFAs in Host Homeostasis

Most produced SCFAs are quickly absorbed by colonocytes whereas only a small amount of these metabolites is excreted in stools. SCFAs lower the luminal pH and alter microbiota in the colon, favoring the growth of butyrate-producing bacteria. SCFAs, in particular butyrate, represent a significant energy source for the host colonocytes and also consume oxygen, contributing to anaerobic conditions [38]. In fact, butyrate and propionate are more efficient at lower concentrations than acetate and lactate and all have a dose-dependent capacity to modulate proinflammatory activation of epithelial and myeloid cells [39].

SCFAs are ligands of G-protein-coupled receptor (GPR) 41 and 43, which are expressed on intestinal, skeletal muscle, liver, and pancreatic tissues (Figure 1). Interestingly, these receptors, also called free fatty acid 3 and 2 (FFA3 and FFA2) receptors, are also expressed in white adipose tissue and mediate SCFA-stimulated leptin secretion. This suggests a distant contribution of the gut microbiota to the host physiology. Acetate mainly binds to GPR43 (FFA2) whereas butyrate binds to GPR41 (FFA3). Propionate is able to bind to both GPR41 and GPR43 [40]. The binding of SCFA to these receptors might be toned down during obesity, promoting hepatic lipogenesis and an energy imbalance [41,42].

SCFAs may contribute to the regulation of host homeostasis via different mechanisms. Indeed, experimental studies in both obese human and animal subjects suggest that acetate and propionate could increase fat oxidation and energy expenditure and decrease lipolysis. Moreover, acetate also contributes to insulin sensitivity through effects on lipid metabolism and glucose homeostasis [43]. Likewise, butyrate is also known to improve insulin sensitivity and intestinal barrier function [43,44]. Propionate stimulates leptin secretion by adipocytes from wild-type but not GPR41 knockout mice [45,46]. Similarly, acetate, but not butyrate, promotes leptin secretion in wild-type mesenteric adipocytes [45]. In addition to the release of leptin, SCFAs can also induce satiety through the modulation of other anorexigenic hormones. For instance, the release of glucagon-like peptide-1 (GLP-1), an anorexigenic incretin produced by enteroendocrine L cells, is stimulated by butyrate [47,48]. Like GLP-1, peptide YY (PYY) is also produced by intestinal L cells through SCFAs’ activation of GPR41 and 43 and is mainly released in the postprandial phase, contributing to the satiety process [49]. Lower plasma GLP-1 concentrations have been reported in obese patients compared to healthy individuals [50,51]. Likewise, obese patients produce less PYY than lean individuals [52]. This incretin impairment in obesity and its association with T2D has been associated with gut dysbiosis [53]. On the other hand, acetate can also reach the brain and act directly on hypothalamic circuitries, inducing the mRNA expression of α-melanocyte-stimulating hormone (α-MSH, an anorexigenic neuropeptide) and reducing the mRNA expression of agouti-related peptide (AgRP, an orexigenic neuropeptide) [54]. Finally, the plasma concentration of the orexigenic hormone ghrelin was positively correlated with the abundance of Bacteroides and Prevotella levels, which are both increased in the microbiota of obese patients [55], and negatively correlated with the abundance of Bifidobacterium and Lactobacillus [56].

Further investigations into the metabolic role of SCFAs are needed to resolve the paradox between the high fecal content of SCFAs and the lack of efficient satiety signaling in obesity. For instance, it has been suggested that the binding of SCFAs to their receptors might be toned down during obesity, promoting hepatic lipogenesis and an energy imbalance [41,42]. It is also worth noting that increased fecal SCFA concentrations could be indicative of greater SCFAs production and/or lower SCFAs absorption [57], and thus may not reflect a full availability for the host [58]. For example, total plasma concentrations of SCFAs were significantly lower in obese diabetic than in control mice [59,60]. In addition, unlike circulating SCFAs, one study found that fecal SCFAs concentrations were actually not associated with insulin sensitivity, lipolysis, or GLP-1 concentrations in humans [61]. According to a recent meta-analysis, “obese individuals had significantly higher SCFA concentrations of acetate (standardized mean difference (SMD = 0.87, 95% CI = 0.24–1.50) in the blood and feces, propionate (SMD = 0.86, 95% CI = 0.35–1.36) in feces and butyrate (SMD = 0.78, 95% CI = 0.29–1.27) in feces than the non-obese subjects”. As explained previously, circulating acetate promotes anorexigenic effects partly through the release of leptin from mesenteric adipocytes. However, while leptin concentrations are elevated in obese patients, the efficacy of the anorexigenic effect of leptin is decreased during obesity (leptin resistance) [62,63]. Another study suggested that dietary non-digestible carbohydrates promote L cell differentiation in the colon of rats [64]. Therefore, it is tempting to speculate that very low dietary fiber (i.e., a Western diet) would decrease the presence of L cells in the colon, downregulating the release of GLP-1 and PYY. Altogether, this highlights a putative dysregulation of SCFA signaling in obesity.

### 2.4. Involvement of Gut Microbiota in Obesity-Associated Low-Grade Inflammation

Low-grade inflammation has been associated with the pathophysiology of obesity and its related complications, such as insulin resistance and resulting cardiovascular disease [65] and T2D [66]. Gut dysbiosis can promote low-grade inflammation through different mechanisms. First, the release by Gram-negative bacteria of lipopolysaccharide (LPS), which can cross the intestinal epithelium through disrupted tight junctions or inside chylomicrons (Figure 1), has been well documented [67]. Changes in the gut microbial composition in obesity, and with a high-fat diet (HFD), also alter the structures of tight-junction proteins [68], leading to enhanced passage of LPS [67]. In the systemic circulation, LPS binds to the LPS-binding protein (LBP) and this complex activates the CD14 receptor. The latter binds to toll-like receptor 4 (TLR4) on macrophages in different organs, such as adipose tissue and liver. As a consequence of this gut microbial TLR4 activation (see Figure 1), genes encoding proinflammatory actors (factor nuclear Kappa B; NF-κB) are strongly expressed, resulting in macrophage infiltration in adipose tissue [69]. LPS infusion in mice reproduces features of HFD-fed mice regarding weight gain, visceral and subcutaneous adiposity, higher fasting glycemia, insulinemia, liver triglyceride content, and body weight as compared to saline-infused mice [67].

### 2.5. Gut Microbial Regulation of Fat Deposition

In addition to their effect on satiety, SCFAs also increase the expression of peroxisome proliferator-activated receptors (PPARs), which are key mediators of adipogenesis [70]. Other studies have shown that butyrate and propionate but not acetate increase the rate of lipolysis in vitro via inhibition of histone deacetylase (HDAC) [71]. Moreover, gut microbiota seem to have a strong influence on fatty acid oxidation. Indeed, germ-free mice fed an HFD showed greater levels of phosphorylated adenosine monophosphate kinase (AMPK) in the liver and skeletal muscles compared to conventionally raised mice fed an HFD [72]. Moreover, these germ-free mice gained significantly less weight than conventionalized mice [72]. In fact, AMPK is a crucial enzyme, which plays an important role in energy homeostasis. Increased levels of AMPK result in stronger fatty acid oxidation [72,73], and the inhibition of this enzyme by gut microbiota promotes cholesterol and triglycerides synthesis, favoring lipogenesis and leading to obesity through excess fat storage [74]. O’Neill and colleagues observed in a large-scale cohort that obesity is associated with reduced AMPK activity without alteration of AMPK expression [75]. It is tempting to speculate that a specific “obesogenic” gut microbiota profile could have a suppressive effect on AMPK activity, making the host more susceptible to obesity (Figure 1) [76]. Gut microbiota transfer from conventionally raised mice into recipient germ-free mice led to the inhibition of another protein involved in the adiposity process: fasting-induced adipose factor (FIAF), also called angiopoietin-like 4 protein (ANGPTL4), by which the microbiota may limit triglyceride accumulation in adipose tissue [77]. FIAF/ANGPTL4 is produced by adipose tissue, liver, skeletal muscle, and the gut in response to fasting. The main role of FIAF/ANGPTL4 is the inhibition of lipoprotein lipase (LPL), leading to a decrease in triglycerides’ accumulation in adipocytes [77]. Conversely, the inhibition of intestinal FIAF promotes fatty acid uptake via increased LPL activity. However, the final effects of different microbiota profiles on the stimulation or inhibition of FIAF remains a matter of debate [78] and needs to be further investigated to confirm the relevance of this pathway for the regulation of fat storage by gut microbiota.

### 2.6. Influence of Gut Microbiota on Bile Acids Metabolism

Bile acids secreted by the hepatocytes play a key role in the digestion and absorption of fatty acids in the small intestine, among other functions. Cholic acid (CA) and chenodeoxycholic acid (CDCA) are the two main primary bile acids produced by hepatocytes from the metabolism of cholesterol and excreted in the bile as conjugates with taurine or glycine [79]. In the intestinal lumen, CA and CDCA are converted by the gut microbiota through deconjugation, dehydrogenation, and dihydroxylation, leading to deoxycholic acid and lithocholic acid, respectively. These secondary bile acids are, further down in the ileum, re-absorbed via both active and passive diffusion and recirculate to the liver via the portal vein [79]. Thus, alteration of the gut microbial composition is likely to influence the bile acid pool and distribution different types. Accordingly, Swann et al. reported that the distinct gut microbiota profile in mice influenced the bile acids profile and energy metabolism [80]. In addition, bile acids can bind to the nuclear Farnesoid X receptor (FXR), which is involved in hepatic lipid and glucose metabolism [81] and hence promotes metabolic liver dysfunction, leading to obesity, insulin resistance (see Figure 1), and nonalcoholic fatty liver disease (NAFLD). Indeed, gut microbiota transfer from HFD-fed mice into recipient germ-free mice led to the development of NAFLD, with hepatic lipid levels similar to that in donor mice [82]. Bile acids are also able to activate G-protein-coupled bile acid receptor 1 (GPBAR1), also known as TGR5, which is abundantly expressed in the intestine, particularly in the ileum and colon. The activation of TGR5 contributes to glucose homeostasis through the release of GLP-1 [83]. Interestingly, modified bile acid profiles have been observed in patients suffering from obesity and T2D and NAFLD [79]. In general, higher plasma bile acid concentrations appear to be positively correlated with obesity, T2D, and NAFLD [79]. Gut microbiota may also influence bile acid metabolism and finally host metabolism (thermogenesis) through additional signaling mechanisms (including GLP-1 expression or enzyme regulation) and finally contribute to increased risk of obesity [84].

Analysis of the gut microbiota has revealed dysbiosis in several pathologies, including obesity, stimulating an interest in the development of therapeutic strategies based on re-shaping of the intestinal microbiota. Thus, traditional probiotics, so-called first-generation probiotics, have been proposed based on the modifications observed in the gut microbial composition between patients and healthy volunteers and generally they come from fermented food. In this context, many naturally beneficial bacteria have been identified as probiotic candidates. More recently, new probiotic candidates, called next-generation probiotics, have been highlighted thanks to new technologies identifying specific alterations during obesity or through mechanistic studies establishing mechanisms of action beneficial for food intake regulation or insulin sensitivity.

## 3. Next-Generation Probiotics as Therapeutic Perspectives in Obesity

### 3.1. Traditional Probiotics

Probiotics are microorganisms that, when administered in an appropriate amount, improve the health of the host [85]. Several probiotic strains, used alone or as bacterial consortium, have been proposed to confer antiobesity effects. The main probiotics proposed so far for obesity are Lactobacillus (*L. casei*, *L. gasseri*, *L. plantarum*, *L. rhamnosus*) and Bifidobacterium (*B. infantis* and *B. longum* for instance) species (Appendix A). These species display low pathogenicity and low levels of antibiotic gene resistance [86]. Basically, probiotic supplementations are used to modulate dysbiotic gut microbiota. Their expected mechanisms of action include a reset of the gut microbial dysregulation previously described, such as the production of bioactive compounds, reduction of fat storage, promotion of fatty acid oxidation, TLRs interaction, reduction of low-grade inflammation, and stimulation of satiety pathways [76,87,88]. In addition to some positive pre-clinical results, clinical evidence has been published for some, but not all, products to support the beneficial effects of the consumption of these traditional probiotics on BMI reduction in obese patients (Appendix A). However, the clinical efficacy remains to be strengthened via repeated clinical studies [89,90], and so far, no single strain or consortium product has been consensually recommended for the treatment of obesity or type2 diabetes [91]. Therefore, the modulation of gut microbiota by traditional probiotics may be helpful as part of a multimodal strategy to improve host metabolic health during obesity but cannot be considered as a first-line treatment.

### 3.2. Next-Generation Probiotics

While traditional probiotics generally show marginal beneficial effects, next-generation probiotics (NGP) have emerged as new preventive and therapeutic tools. As explained by O’Tool et al., NGP is a “well-characterized probiotic strain which could be used as delivery vehicles for a specific molecule abrogating the disease phenotype and thus promoting health” [92]. The authors also suggest that the “term NGP is a reasonable attempt to mark the progression from traditional microorganisms with long histories of safe use to untried microorganisms with no such historical acceptance”. Recent studies (described below) have unraveled several NGP whose efficacy is supported by pre-clinical and clinical evidence as reported in Table 1.

The *A. muciniphila* strain, which is depleted in the microbiota of obese patients, has been identified as a probiotic candidate for the care of patients with insulin resistance, a common complication of obesity. Indeed, administration of *A. muciniphila* (daily 5 × 10^9^ cfu/mL) in animal models of obesity, such as HFD mice, reduced insulin sensitivity, fat deposition, and weight gain [93,94]. More specifically, *A. muciniphila* supplementation, alive or pasteurized (10^10^ bacteria per day), improved some metabolic parameters associated with obesity, such as insulin resistance, hepatic steatosis, and intestinal permeability [93]. In a pilot clinical study, *A. muciniphila* supplementation (10^10^ bacteria per day) in overweight/obese subjects improved insulin sensitivity and reduced insulinemia and plasma total cholesterol [93]. However, only a trend toward a decrease in body weight, fat mass, and hip circumference was observed in this study [93]. The molecular mechanism of action of *A. muciniphila* on insulin resistance has been identified recently. Indeed, a protein purified from *A. muciniphila* (called “P9”) and administered to mice induced GLP-1 release, improved glucose homeostasis, and activated brown adipocytes, promoting thermogenesis [98].

A former study underlined the interaction of Amuc_1100, a specific protein isolated from the outer membrane of *A. muciniphila*, with toll-like receptor 2, suggesting the role of *A. muciniphila* in the regulation of the NF-κB pathway [99]. In addition, prebiotics, such as the inulin-type fructans, increase the abundance of *A. muciniphila*, which may contribute to the improvement of metabolic disorders and obesity by prebiotics [100]. Thus, although *A. muciniphila* appears to be a promising candidate for improving insulin resistance, its efficacy in obesity remains to be demonstrated. Its clinical usefulness in established diabetes may be challenged by pharmacological approaches, including GLP-1 agonists. So, the positioning of *A. muciniphila* may rather be for the early stages of insulin resistance.

Recently, *Dysosmobacter welbionis* (*D. welbionis*) was also identified as having beneficial effects similar to those of *A. muciniphila* [101]. In fact, supplementation with live *D. welbionis J115T* (1.0 × 10^9^ daily) in mice, but not with the same pasteurized strain, induced a decrease in fat mass gain, insulin resistance, and white adipose tissue hypertrophy and inflammation. In addition, this strain’s administration protected the mice from brown adipose tissue inflammation in association with an increased number of mitochondria and thermogenesis [101].

*Christensenella minuta* (*C. minuta*), a bacteria from the Clostridia class, is a novel interesting live bio-therapeutic product since its administration prevented weight gain and toned down glycemia and plasma leptin in diet-induced obese mice (2 × 10^9^ daily) [95]. Moreover, *C. minuta* has been shown to lower hepatic triglycerides and free fatty acids accumulation, prevent adipogenesis, maintain gut epithelium integrity in vitro, and modulate the fecal microbiome [95]. In addition, an increased abundance of Christensenellaceae was associated with low BMI [100]. Moreover, in the humanized SHIME^®^ model, chronic daily administration of the strain *C. minuta DSM33407* (2 × 10^9^) significantly improved microbial diversity in the distal colon, increased SCFAs concentrations, and restored the Firmicutes/Bacteroidetes ratio. Thus, the administration of some new-generation probiotic strains could modify the composition of the gut microbiota, promoting the production of SCFAs and possibly their absorption.

The three next-generation probiotics described above mainly have metabolic effects but no demonstrated clinical efficacy on obesity, and their possible impact on the regulation of food intake has not been evaluated. As a matter of fact, hyperphagia, or any kind of compulsive eating disorder, is a common factor in the constitution of obesity, and most obese patients report reduced perception of satiation [102]. Thus, addressing the regulation of food intake via modulation of the microbiota should be considered as a logical target for early intervention before the onset of severe obesity with resulting metabolic complications.

Reducing ghrelin orexigenic signaling could be a target for probiotic interventions. In fact, previous studies have reported a negative association between serum ghrelin levels and the quantity of Bifidobacterium and Lactobacillus, which may favor weight gain [56]. However, data in the literature regarding modulation of ghrelin signaling by probiotics are still missing.

Satiation is induced rapidly during and after meal consumption via several neuro-hormonal pathways, including the release of GLP-1 and PYY as inducers of the final activation of α-MSH anorexigenic signaling within the hypothalamic regulation network, leading to the cessation of eating and conditioning the interval until the next meal [103]. The probiotic strain *Hafnia alvei 4597* (*H. alvei 4597*) currently seems most promising for the limitation of excess weight gain by promoting the satiation pathway both in animals and humans. The mechanisms of action of *H. alvei 4597* include increased production of the caseinolytic protease B (ClpB) protein, which has been identified as a conformational mimetic of α-MSH [104]. In previous experiments, nutrient-fed E. coli produced increased amounts of ClpB, which reduced food intake in normal rats and activated central anorexigenic pathways [105]. In addition, ClpB increased PYY in vitro release by colonic cells [106]. Treatment with H. alvei decreased body weight gain and fat mass gain and reduced food intake in HFD-fed and ob/ob hyperphagic mice [96,107]. These effects were associated with reduced hyperglycemia, plasma total cholesterol, and alanine aminotransferase, suggesting an associated improvement in the metabolic consequences of these obesogenic conditions. Finally, in a recent double-blind placebo-controlled study in overweight volunteers, supplementation with *H. alvei 4597* (100 billion bacteria daily), in addition to standard dietary counselling, promoted improved weight loss, increased the feeling of fullness, and resulted in a greater loss of hip circumference and reduction of fasting glycemia [97].

Some other next-generation probiotics have recently been proposed [100]. *Prevotella copri* reduces insulin resistance in animals; *Bacteroides thetaiotaomicron* has displayed strong efficacy in preclinical models of inflammatory bowel disease, protecting against weight loss and histopathological changes in the colon and inflammatory markers. Moreover, *Faecalibacterium prausnitzii*, which has anti-inflammatory actions at the colonic level, may also reduce obesity-related micro-inflammation. Finally, oral administration of live, but not heat-killed, *Parabacteroides goldsteinii* bacteria to HFD-fed mice considerably reduced weight gain and obesity-associated metabolic disorders [100].

Fecal microbiota transplantation may be the ultimate approach to treat obesity. Indeed, pilot studies have reported that transplantation from lean donors to patients with metabolic syndrome improved insulin sensitivity [108]. This effect seems to be related to the reduction of chronic low-grade inflammation [109]. However, data on the benefits/safety balance of FMT in obesity are still insufficient and this method should not be considered unless in controlled studies [110].

## 4. Summary and Conclusions

Obesity is a worldwide health concern that is continuing to rise rapidly. Obesity is a multifactorial disease and the gut microbiome has recently emerged as a key factor in its pathophysiology. Gut microbiota may favor obesity by various mechanisms, including the control of energy homeostasis, LPS-stimulated inflammation, bile acids, and the regulation of fat deposition. Regarding SCFAs, their effect on food intake during obesity remains to be discussed and mechanisms of action in obese patients must be further clarified. However, until now, it has been unclear which specific bacterial community contributes to the development of obesity. Already, modulation of the gut microbiota appears to be a powerful contribution to the multimodal care of obesity. However, additional clinical and mechanistic studies are needed to further support the clinical usefulness and safety of microbiota modulation by next-generation probiotics in obesity.

## Figures and Tables

**Figure 1 microorganisms-10-00452-f001:**
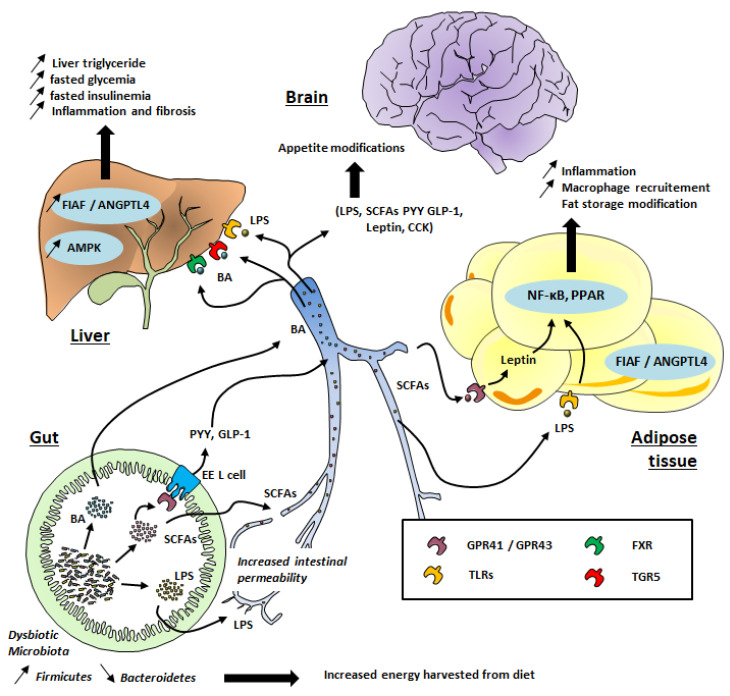
Gut microbiota’s contribution to obesity development. TLR: Toll-Like Receptor; GPR: G-protein coupled receptor; FXR: Farnesoid X Receptor; TGR5: Takeda G protein-coupled receptor 5; SCFAs: Short-Chain Fatty Acids; LPS: lipopolysaccharides; PPAR: Peroxisome proliferator-activated receptors; FIAF: Fasting-induced adipose factor; ANGPTL; angiopoietin-like protein; BA: Bile Acid, EE, Entero-endocrine cells; CCK: Cholecystokinin; PYY: peptide tyrosine tyrosine; GLP-1: Glucagon-like peptide 1; AMPK: AMP-activated protein kinase, NF-κB: Nuclear factor-*kappa B*.

**Table 1 microorganisms-10-00452-t001:** Next-generation probiotics’ efficacy in the care of obesity in animal and human clinical trials.

	Pre-Clinical Evidence					Clinical Evidence					
Composition	Weight gain	Food Intake	Fat accumulation	Inflammation	Comorbidity	Weight gain	Food Intake	Fat accumulation	Inflammation	Comorbidity	References
*Akkermansia Muniniphila*	↓	=	↓	↓	↓ Total cholesterol ↑ Glucose tolerance	↓ *		↓ *	↓	↓ Insulin resistance ↓ Insulinemia ↓ Total cholesterol	[93,94]
*Christensenella minuta DSM33407*	↓	=	↓	↓	↓ Glycemia ↓ Leptin (leptin resistance?)						[95]
*Hafnia Alvei 4597*	↓	↓	↓ *		↓ * Total cholesterol ↓ ALAT	↓	↑ feeling of fullness			↓ Hip circumference ↓ Fasting glycemia	[96,97]
*Dysosmobacter welbionis J115T* (live)	↓ *		↓	↓	↑ Glucose tolerance ↓ * Insulin resistance ↓ Leptin						[98]

* Trend.

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
