# Peer review of "Dysbiotic Gut Bacteria in Obesity: An Overview of the Metabolic Mechanisms and Therapeutic Perspectives of Next-Generation Probiotics"

_microorganisms, 2022, doi:10.3390/microorganisms10020452_

Round 1

Reviewer 1 Report

I thank the authors for paying attention to this topic which is really useful and of great importance for the scientific community. However, none methodological approach was used to perform the review and is need to revision the manuscript adding the methods of reviewed and also the innovation that this articles contributes to the scientific community. Moreover, the specific focus on different variables should be considered such as diet, physical activity and lifestyles and analysing the results of review in this way.
Some minor revisions are need and suggested:
- all acronyms should be expanded in the text and in the figure and table. 
- Table 1 not is very clear and should be implemented and explained in clear way.

Reviewer 2 Report

 Present review paper "Dysbiotic gut bacteria in obesity: overview of metabolic mechanisms and therapeutic perspectives" is devoted to the important subject - description of the role of microbiota in obesity. I am sure that the subject deserves discussion and careful description. I think that the authors rave done their job in describing what was written regarding microbiota and obesity very well. The language is clear and easy read. There are minor mistakes in the text such as “E.Coli” (l.301), or no brackets on (l. 312).

What I missed was a discussion showing position of the authors regarding several things/ For example SCFA can do different things inducing or impeding the obesity:

" Experimental studies in both obese human and animal subjects suggested that an increased production of SCFAs could provide additional calories to the host and thus, could lead to weight gain".

 "SCFAs can also induce satiety through the release of 
other anorexigenic hormones"

In this respect do we want SCFA producers in the gut or no?

The same is related to the Firmicutes vs Bacteroidetes ratio. How valuable is this criterion when many of the anti obesity New generation probiotics are the firmicutes? And 90% of entire microbiota content are Firmicutes + Bacteroidetes?

In short I would suggest expanding the conclusion part which can show the position of the authors, not just careful making a school task.

Reviewer 3 Report

Evaluation manuscript: microorganisms-1533505

Title: Dysbiotic gut bacteria in obesity: overview of metabolic mechanisms
and therapeutic perspectives

Article Type: Review

Author: Jonathan Breton, Marie Galmiche *, Pierre Déchelotte

Content of the paper

Breton et al. reviewed the current information about the correlation between the gut microbiome and obesity. Thereby, they focused on metabolic mechanisms, such as energy homoestasis, low grade inflammation, microbial metabolites, LPS and bile acid metabolism and therapeutic perspectives.

Comments concerning the paper

1.) The manuscript is generally well-written and the presented information straight forward. I added some minor comments, remarks, question which should be incorporated or at least answered.

Originality

The study gives a good overview about the correlation of the gut microbiome and obesity. However, there are also some other reviews about this topic (e.g., first two google hits: https://pubmed.ncbi.nlm.nih.gov/27255389/, https://www.ncbi.nlm.nih.gov/pmc/articles/PMC7333005/) and the authors should refer them and state the benefit of their own review.

Structure and Language

The text is well written and well structured.

Strength and Limitations

  • Strength The review gives a good overview of the topic
  • Limitation: Not completely convinced about its value compared to other reviews with a similar topic.

Title

  • In principle, the title is fine. However I have the philosophic question: What is then as a “healthy”/ “Stable” microbiome. Would be really worth to a add chapter to link back what we want to achieve then with probiotics.

Abstract

  • fine

Introduction

  • Line 23: please be precise and state the prevalence of obesity and how much it is rising.

Material and method

  • Please add a small paragraph about your type of review (https://libguides.csu.edu.au/review/Types) and how you have performed it/ decided what you included.

Results

  • Line 39: You start with obesity-associated dysbiosis. However, it would be useful to define what is a “normal” microbiome
  • Also it would be useful for the readers to give a short overview about the methods usually applied for assessing the microbiome starting including metagenomics, transcriptomics, metaproteomics (e.g,. https://www.mdpi.com/2218-273X/11/5/726), and metabolomics as well as the sampling method (feces, vs. biopsis). At this stage or the end, it would be good to discuss how much the microbiome is involved or just co-correlating to the diet.
  • Line 42: species names are usually written italic.
  • Line 46: write “Archaea” in capital letters
  • Line 55-59: For the ratio between Firmicutes and Bacteriodetes, I would be carefully since this is not found in all studies (Mariat, D.; Firmesse, O.; Levenez, F.; Guimarăes, V.; Sokol, H.; Doré, J.; Furet, J.P. The Firmicutes/Bacteroidetes ratio of the human microbiota changes with age. BMC Microbiol. 2009, 9, 123. [Google Scholar] [CrossRef]). Also to define it with a ratio of “1” is difficult since this is based on 16S. With other omics methods this may be different. For example in our metaproteomics study about weight loss we could also not confirm it (https://www.mdpi.com/2218-273X/11/5/726/htm#B35-biomolecules-11-00726). à At this point I would recommand you to discuss this more carefully.
  • Line 93: capital letters “Bifidobacteria”
  • Line 100 “Production of short chain fatty acid”: In this chapter it would be good to discuss the impact of the microbiome to weight gain based on some numbers, targeting the question how much extra calories could be provided by the microbiome. So how much SCFA are produced by 40-60 g fibres, how much SCFA are up taken by the human and to how much calories this equals?
  • Chapter 2.3. In this chapter it would be useful, to elaborate that the butyrate is used in colon cells to produced energy and consume oxygen which contributes to anaerobic conditions. You may consider in this chapter also Lactate (e.g., https://www.sciencedirect.com/science/article/pii/S0171298515300085)
  • Line 198, Space after comma
  • Line 201, Space after NAFLD
  • Line 244: How you define a next generation probiotics?
  • Chapter 3.2: Would be useful to know the actual amounts of the probiotics required to observe the effect?
  • Line 301: Writing E.Coli (small “coli”)
  • Line 312 unknown “73”=?
  • You focus the review on bacteria, what is with Yeast, Bacteriophages, or Archaea?
  • It would be interesting to illuminate the interplay between human and microbial hydrolysis enzymes for food degradation
  • Another missed aspect is the process view. Obese people usually eat more food, thus this impacts the time food stays in the gut. If the transfer through the gut takes more time, microbes have more time to growth…how does this impacts the microbiome?
  • Fermentative microbial processes produce gas (CO2 or methanes by Archaea). Is there something known about this and obesity
  • As far as I know antibiotics have a big impact on the gut microbiome (https://www.ncbi.nlm.nih.gov/pmc/articles/PMC6287021/) and on obesity, which could be added to your review.
  • You may also elaborate about the interaction between mucus, the microbiome and obesity.
  • You restrict the therapeutic options only to probiotics. What is with other strategies to impact the microbiome such as prebiotics, bacteriophages, or antibiotics?

Conclusion

  • is ok

Suggestion to the editor

After in death evaluation of the presented manuscript, the reviewer would like to recommend the editor:

b.) to accept this paper under consideration of following minor modification.
